# Cost-effectiveness of HPV-based cervical cancer screening in the public health system in Nicaragua

Nicole G Campos,[1] Mercy Mvundura,[2] Jose Jeronimo,[3] Francesca Holme,[3] Elisabeth Vodicka,[4] Jane J Kim[1]

► Prepublication history and additional material are available. To view, please visit the journal (http://dx.doi.org/10.1136/bmjopen-2016-015048).

[1]Center for Health Decision Science, Harvard T.H. Chan School of Public Health, Boston, Massachusetts, USA
[2]PATH, Devices and Tools Program, Seattle, Washington, USA
[3]PATH, Reproductive Health Global Program, Seattle, Washington, USA
[4]University of Washington, School of Pharmacy, Seattle, Washington, USA

**Correspondence to**
Dr Nicole G Campos;
ncampos@hsph.harvard.edu

## ABSTRACT

**Objectives** To evaluate the cost-effectiveness of human papillomavirus (HPV) DNA testing (versus Papanicolaou (Pap)-based screening) for cervical cancer screening in Nicaragua.

**Design** A previously developed Monte Carlo simulation model of the natural history of HPV infection and cervical cancer was calibrated to epidemiological data from Nicaragua. Cost data inputs were derived using a micro-costing approach in Carazo, Chontales and Chinandega departments; test performance data were from a demonstration project in Masaya department.

**Setting** Nicaragua's public health sector facilities.

**Participants** Women aged 30–59 years.

**Interventions** Screening strategies included (1) Pap testing every 3 years, with referral to colposcopy for women with an atypical squamous cells of undetermined significance or worse result ('Pap'); (2) HPV testing every 5 years, with referral to cryotherapy for HPV-positive eligible women (HPV cryotherapy or 'HPV-Cryo'); (3) HPV testing every 5 years, with referral to triage with visual inspection with acetic acid (VIA) for HPV-positive women ('HPV-VIA'); and (4) HPV testing every 5 years, with referral to Pap testing for HPV-positive women ('HPV-Pap').

**Outcome measures** Reduction in lifetime risk of cancer and incremental cost-effectiveness ratios (ICER; 2015 US$ per year of life saved (YLS)).

**Results** HPV-based screening strategies were more effective than Pap testing. HPV-Cryo was the least costly and most effective strategy, reducing lifetime cancer risk by 29.5% and outperforming HPV-VIA, HPV-Pap and Pap only, which reduced cancer risk by 19.4%, 12.2% and 10.8%, respectively. With an ICER of US$320/YLS, HPV-Cryo every 5 years would be very cost-effective using a threshold based on Nicaragua's per capita gross domestic product of US$2090. Findings were robust across sensitivity analyses on test performance, coverage, compliance and cost parameters.

**Conclusions** HPV testing is very cost-effective compared with Pap testing in Nicaragua, due to higher test sensitivity and the relatively lower number of visits required. Increasing compliance with recommended follow-up will further improve the health benefits and value for public health dollars.

### Strengths and limitations of this study

► Strengths of this study include the use of implementation data from the Scale-Up project to estimate the cost-effectiveness of human papillomavirus testing in Nicaragua's public health system. Findings were robust across extensive sensitivity and scenario analyses.

► A limitation of this study is that modelled screening algorithms do not entirely reflect the complex downstream management described by in-country guidelines. While the modelled algorithms reflect the prototypical structure of a screening episode and the type of facility at which visits usually take place, these do not capture variation due to geography or health facility capacity.

► An additional limitation is that while we adopted a micro-costing approach to leverage data from implementation in Nicaragua, individual-level data for each woman were not available; furthermore, we did not have cost data associated with HPV self-collection in community settings, where most self-collection takes place.

estimated 934 cases and 424 deaths each year.[1] Yet, cervical cancer is preventable through screening that allows for early detection and subsequent treatment of precancerous lesions caused by sexually transmitted infection with human papillomavirus (HPV). While most HPV infections clear spontaneously within 1 to 2 years, a persistent infection with one of approximately 15 oncogenic HPV genotypes may progress to precancer which, if untreated, may become invasive cancer.[2 3]

In most high-income countries, routine screening with cervical cytology (ie, Papanicolaou (Pap)) testing has substantially reduced the incidence of cervical cancer.[4] However, due to its low sensitivity to detect precancer, Pap testing must be performed at regular frequent intervals in women of screening age. In low-income and middle-income settings, where many women do not have access to routine primary healthcare

## BACKGROUND

Cervical cancer is the leading cause of cancer among women in Nicaragua, with an

and even fewer have access to higher-level facilities that offer diagnostic testing and treatment, Pap testing has not been effective at reducing cervical cancer incidence and mortality.[5] In Nicaragua, an estimated 31.5% of women aged 15 to 49 years have been screened within the last year, and nearly 30% of women in this age group have never been screened.[6] One recent survey found that 87% of women in León, Nicaragua, were informed of their Pap results, but of those who were referred to follow-up, only 67% received further care.[7]

HPV DNA tests are highly sensitive to detect potentially oncogenic HPV infections and present an alternative to Pap-based screening. Because HPV-negative women are at very low risk for developing cervical cancer within the next 10 years,[8] the interval between screenings can be extended to at least 5 years for this subset of women.[9] An additional advantage of HPV testing is that samples can be collected by a provider or by the woman herself, reducing the burden on health workers and time women spend seeking care and potentially increasing screening uptake.[10–13] Furthermore, a lower-cost HPV DNA test known as careHPV has been clinically validated[14 15] and is now commercialised. Given these potential benefits, the WHO recommends HPV testing for countries with sufficient resources.[9]

In 2011, the Screening Technologies to Advance Rapid Testing for Cervical Cancer Prevention–Utility and Program Planning (START-UP) project in Nicaragua's Masaya department demonstrated that screening with careHPV could be effectively implemented in public sector health facilities.[14] The Ministry of Health of Nicaragua subsequently built on these initial efforts, incorporating HPV testing into public healthcare systems in three departments with technical assistance from PATH under the Scale-Up project. Adoption of HPV testing within Nicaragua's public healthcare system is taking place in three phases.[16] In phase 1, partner organisations worked with the Ministry of Health to prepare for introduction of HPV screening into public health facilities by developing screening and treatment algorithms, creating educational materials, organising training sessions for health workers and laboratory technicians and bolstering referral and treatment systems for follow-up of screen-positive women. Phase 2 piloted screening with 10 000 HPV tests in order to identify and address barriers to implementation. Phase 3 will expand coverage to over 50 000 women within 1 year.

To inform decision makers considering the national adoption and scale up of HPV testing within Nicaragua's public health sector, this study aimed to (1) estimate the economic cost of cervical cancer screening with careHPV testing and (2) project the long-term health and economic impact and value (ie, cost-effectiveness) of careHPV testing in Nicaragua relative to existing Pap-based screening.

## METHODS

### Analytic overview

We used a micro-costing approach to measure and aggregate the cost of all resources used to provide cervical cancer screening at the level of the individual patient within the public health sector in Nicaragua. We considered direct medical costs (ie, medical resources required for the intervention), direct non-medical costs (ie, other resources consumed as part of the intervention, such as patient transportation costs) and patient time costs (ie, time spent travelling and waiting for or receiving care). These cost data were input into a previously developed Monte Carlo simulation model (programmed in C++) of the natural history of HPV infection and cervical cancer that was calibrated to epidemiological data from Nicaragua.[17 18] We then used the model to project the lifetime health and economic outcomes associated with careHPV testing, using three different algorithms for the management of women who test HPV positive and Pap-based screening for women aged 30 to 59 years.

Model outcomes included the lifetime risk of cervical cancer, total lifetime costs per woman (in 2015 US$) and life expectancy. Incremental cost-effectiveness ratios (ICER) were calculated by dividing the additional cost of a particular strategy by its additional health benefit, compared with the next most costly strategy. Dominated strategies (defined as more costly and either less effective or having a higher incremental cost-effectiveness ratio than more effective strategies) were eliminated. There is no universal criterion that defines a threshold cost-effectiveness ratio, below which an intervention is considered good value for money; we considered an intervention with an ICER less than Nicaragua's 2015 per capita gross domestic product (GDP) of US$2090 to be 'very cost-effective', and an intervention with an ICER less than three times per capita GDP as 'cost-effective'.[19] We followed guidelines for cost-effectiveness by adopting a societal perspective, including costs irrespective of the payer in order to capture the opportunity cost of resources used for the screening intervention. We discounted future costs and life-years at a rate of 3% per year to account for time preferences (Supplementary Data).[20 21]

### Mathematical simulation model

Descriptions of the natural history model of HPV infection and cervical carcinogenesis and model parameterisation process have been previously published,[17 18] but we summarise model features here. Individual girls enter the model at age 9 years, prior to initiating sexual activity, and face monthly transitions between mutually exclusive health states that reflect disease progression, including type-specific HPV infection, grade of precancer (ie, cervical intraepithelial neoplasia (CIN) grade 2 or 3) and stage of invasive cancer. Transition probabilities may vary by age, HPV type, duration of infection or precancerous lesion status, prior HPV infection and exposure to screening and treatment of HPV or precancer. Cervical cancer can be detected through symptoms or screening.

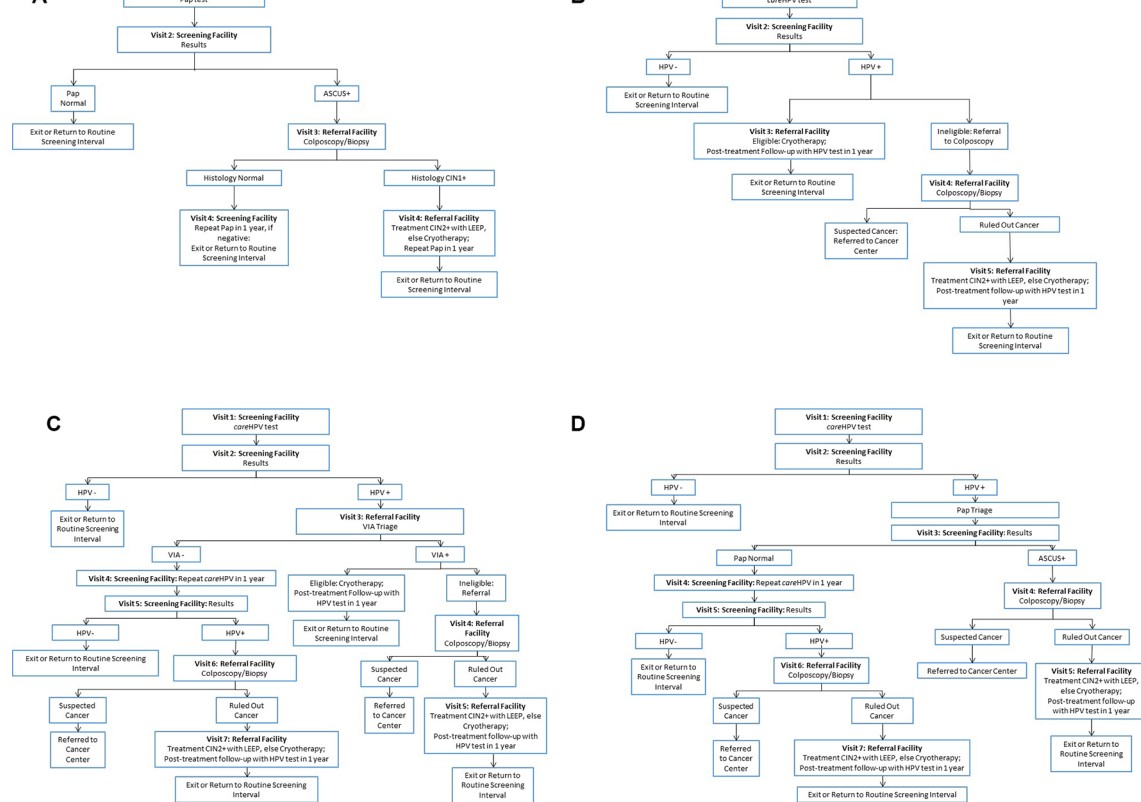

**Figure 1** Pathways of care, by screening strategy. Each diagram indicates the flow of screening-eligible women (ie, women aged 30 to 59 years) through each point of contact in a screening episode, conditional on visit compliance and test results, for (A) Pap testing every 3 years (Pap), which requires four visits for screening, diagnosis and treatment; (B) HPV testing with referral to cryotherapy for all HPV-positive eligible women every 5 years (HPV-Cryo), which requires three or more visits for screening and necessary treatment; (C) HPV testing followed by visual inspection with acetic acid (VIA) triage of HPV-positive women every 5 years (HPV-VIA), which requires three or more visits for screening and necessary treatment; and (D) HPV testing followed by Pap triage of HPV-positive women every 5 years (HPV-Pap), which requires five or more visits for screening and necessary treatment. ASCUS , Pap result of atypical squamous cells of undetermined significance or worse; CIN, cervical intraepithelial neoplasia; HPV, human papillomavirus; LEEP, loop electrosurgical excision procedure; VIA, visual inspection with acetic acid.

Death can occur from non-cervical causes or from cervical cancer after its onset. The model tracks each individual woman's health status, clinical events and economic outcomes over her lifetime and aggregates outcomes to estimate the expected costs and health outcomes over the lifetime of the cohort.

The model was calibrated to epidemiological data on age-specific HPV prevalence and cervical cancer incidence from Nicaragua.[1 14 17] We estimated baseline 'prior' input parameter values for natural history transitions using available longitudinal data, including age- and type-specific HPV incidence data from Colombia.[22–25] To reflect potential differences in parameters that may vary by setting (ie, age-specific and type-specific HPV incidence, natural immunity following initial infection) and uncertainty in progression and regression of precancer, we set plausible bounds around these input values and performed repeated model simulations of disease natural history in the absence of any intervention. Each model simulation selected one random value within the bounds for each uncertain parameter, creating a unique natural history input parameter set. By summing the

log-likelihood of model-projected outcomes for each parameter set relative to the epidemiological data from Nicaragua, we computed a goodness-of-fit score. We selected the 50 top-fitting input parameter sets to use in analysis. Results are reported as the mean across the top 50 parameter sets, and ICER are reported as the ratio of the mean costs divided by the mean effects of one strategy versus another across sets.[26] Further details on model parameterisation, including calibration, are available in the online supplementary appendix.

## Cervical cancer screening strategies
We considered the following screening strategies for women aged 30 to 59 years (figure 1): (1) Pap testing every 3 years, with referral to colposcopy for all women with an atypical squamous cells of undetermined significance or worse result (hereafter referred to as 'Pap'); (2) careHPV testing every 5 years, with referral to cryotherapy for all HPV-positive eligible women (HPV cryotherapy or 'HPV-Cryo'); (3) careHPV testing every 5 years, with referral to triage with visual inspection with acetic acid (VIA) for all HPV-positive women ('HPV-VIA'); and (4)

careHPV testing every 5 years, with referral to Pap testing for all HPV-positive women ('HPV-Pap'). The pathway of care for each strategy was based on patterns of care in the Scale-Up project, national screening guidelines and WHO recommendations. We optimistically assumed 70% of women had access to routine screening and attended an initial visit at a screening facility (ie, a primary health-care facility). Women could then return to receive screening results and recommendations for any necessary follow-up care. Follow-up could include colposcopy, cryotherapy or triage testing at a referral (ie, higher level) facility, with the exception of Pap triage testing, which was assumed to take place at the screening clinic. At each encounter after the initial screening visit, we assumed 85% of women complied with each subsequent visit to a screening facility, while 40% complied with each subsequent visit to a referral facility, consistent with data from Phase 2 of the Scale-Up project. The minimum number of visits required for treatment in a single screening episode was four for Pap, three for HPV-Cryo, three for HPV-VIA and five for HPV-Pap. In the HPV-VIA and HPV-Pap strategies, women who were HPV positive but negative on the selected triage test were referred to repeat HPV testing in 1 year. In the HPV-Cryo and HPV-VIA strategies, women who were not eligible for treatment with cryotherapy based on visual assessment were referred to colposcopy with biopsy to rule out cancer; in the absence of cancer, these women were referred to treatment with loop electrosurgical excision procedure (LEEP) or cryotherapy.

Screening and treatment parameters are presented in table 1.[6 14 17 27–32] Screening test performance data were drawn from the START-UP project in Nicaragua to reflect local test characteristics. While the START-UP project did not evaluate VIA and Pap as triage tests, we used the VIA and Pap positivity rates in HPV-positive women, along with published studies of triage test performance, to inform triage test sensitivity and specificity[33–38]; in the base case, we optimistically assumed high sensitivity of triage testing.

For all HPV testing strategies, we assumed 20% of women received provider collection of cervical specimens and 80% of women self-collected vaginal specimens, consistent with the proportions in the Scale-Up project to date. We weighted cost and health outcomes for provider and self-collection accordingly when aggregating results for the HPV strategies.

## Cost data

All costs were converted to 2015 US$ using GDP deflators and the official exchange rate.[39] The direct medical costs of screening, diagnosis and treatment of precancer were drawn from the START-UP study (Masaya department) and the Scale-Up project (Carazo, Chontales and Chinandega departments). Direct medical costs included clinical staff time, clinical supplies, drugs, clinical equipment, laboratory staff time, laboratory supplies and laboratory equipment. Direct non-medical costs included women's round-trip transportation costs to health facilities and were based on estimates provided by Scale-Up project staff

to represent average transportation costs in the Carazo, Chontales and Chinandega departments. To account for the opportunity cost of women's time spent travelling to, waiting for or receiving care, we used time estimates from the START-UP and Scale-Up projects and valued women's time using Nicaragua's monthly minimum wage to serve as a proxy for the societal value of women's time. Figure 2 displays the categorical breakdown of undiscounted costs for the screening visits over the course of a woman's screening-eligible years, with Pap versus careHPV testing.

Data on programmatic costs are limited, but for HPV strategies, we included the cost of training sessions for health providers offering HPV-based screening, outreach workers, laboratory technicians and providers offering VIA and cryotherapy. While women who self-collected HPV specimens in the Scale-Up project primarily did so in a community setting, micro-costing data were not available for self-collection performed outside of the clinic, so we assumed clinic-based self-collection. However, we conservatively included the cost of training outreach workers to represent a known programmatic cost as self-collection efforts are shifting to community settings.

Data on the costs of treating cervical cancer were unavailable for Nicaragua, so we estimated direct medical, direct non-medical and patient and support person time costs using data from El Salvador.[29]

Selected cost data are presented in table 1. Further details on cost data are provided in the online supplementary appendix.

## Sensitivity analyses

We performed sensitivity analysis to examine the impact of independently varying uncertain parameters, including Pap test performance, triage test performance in HPV-positive women, colposcopy performance, screening coverage, visit compliance, eligibility for cryotherapy following a positive screening and triage test, treatment effectiveness, discount rate and cost data. Ranges selected for sensitivity analysis are displayed in table 1.

## Scenario analysis

The base-case and sensitivity analyses assumed the availability of all strategies (ie, Pap, HPV-Cryo, HPV-VIA and HPV-Pap). Additionally, we performed a scenario analysis in which we assumed HPV-Cryo was not available for logistical and programmatic reasons (ie, limited access to cryotherapy equipment and gas).

## RESULTS
### Base case: population-level health benefits and cost-effectiveness analysis

HPV-based screening strategies were more effective than Pap testing. Among the HPV strategies, HPV-Cryo (every 5 years) was the most effective strategy; under base-case assumptions, it reduced the lifetime risk of cervical cancer by 29.5% on average (range, 25.2%–33.6%). HPV-VIA (every 5 years) reduced cancer risk by 19.4% (range, 16.2%–22.6%), while HPV-Pap reduced cancer

**Table 1** Baseline values and ranges for model variables

| Test/parameter | Base case | Sensitivity analysis |
|---|---|---|
| **Screening/triage test performance (sensitivity/specificity to detect CIN2+)** | | |
| Pap (primary)[14 27 28] | 0.41/0.94 | Alternative sensitivity/specificity pair, 0.70/0.95 |
| CareHPV (primary), provider-collection of cervical samples[14] | 0.78/0.89 | – |
| CareHPV (primary), self-collection of vaginal samples[14] | 0.67/0.86 | – |
| VIA (triage of HPV+)[14 33 34 36 37] | 0.60/0.75 | Alternative sensitivity/specificity pairs, 0.40/0.85 and 0.70/0.65 |
| Pap (triage of HPV+)[14 33 35 38] | 0.85/0.55 | Alternative sensitivity/specificity pairs, 0.40/0.85 and 0.90/0.50 |
| **Colposcopy performance (sensitivity/specificity to detect CIN1+)[14 17]*** | | |
| Colposcopy (ASCUS+ women) | 0.95/0.68 | 1.0/1.0 |
| Colposcopy (HPV+ women ineligible for cryotherapy) | On ruling out cancer, all referred to treatment | 0.95/0.68, with <CIN1 sent back to routine screening intervals |
| Colposcopy (HPV+/VIA+ women ineligible for ST cryotherapy) | On ruling out cancer, all referred to treatment | 0.95/0.68, with <CIN1 sent back to routine screening intervals |
| Colposcopy (HPV+/Pap+ women) | On ruling out cancer, all referred to treatment | 0.95/0.68, with <CIN1 sent back to routine screening intervals |
| **Coverage and compliance†** | | |
| Access to routine screening, % of the target population[6] | 70% | 50%–80% |
| Visit compliance, screening facility‡ | 85% | 40%–85% |
| Visit compliance, referral facility‡ | 40% | 40%–85% |
| **Treatment eligibility and efficacy** | | |
| Eligibility for screen-and-treat cryotherapy[29 42] | ≤CIN1, 100%; CIN2, 85%; CIN3, 75% | ≤CIN1, 75%; CIN2, 60%; CIN3, 49% |
| Screen-and-treat cryotherapy cure rate (for HPV+ or HPV+/VIA+)[30–32] | 92% | 75% |
| Proportion of women maintaining an HPV infection following cryotherapy[43] | 15% | – |
| Treatment cure rate following colposcopy (LEEP for CIN2+; else cryotherapy)[30] | 96% | 85% |
| Proportion of women maintaining an HPV infection following colposcopic diagnosis and treatment[44] | 10% | – |
| Discount rate for costs and life-years | 3% | 0%–5% |

Continued

**Table 1** Continued

| Test/parameter | Base case | Sensitivity analysis |
|---|---|---|
| Direct medical costs, screening and treatment of precancer (2015 US$)[14][‡] | | |
| Pap test | 7.26 | 3 |
| CareHPV test (provider-collection) | 11.96 | – |
| CareHPV test (self-collection) | 11.04 | 75%–125% of base case |
| VIA triage test | 4.19 | – |
| Colposcopy/biopsy | 19.91 | 6.91§ |
| Cryotherapy | 18.16 | 30.54¶ |
| LEEP | 68.36 | 75%–125% of base case |
| Women's time and transportation costs (2015 US$)[‡] | | |
| Transportation to screening facility (round trip) | 0.41 | 0%–50% of base case |
| Transportation to referral facility (round trip) | 2.81 | 0%–50% of base case |
| Wait time, screening facility | 0.48 | 0%–50% of base case |
| Wait time, referral facility | 1.75 | 0%–50% of base case |
| Transport time, screening facility | 0.82 | 0%–50% of base case |
| Transport time, referral facility | 3.10 | 0%–50% of base case |
| Programmatic costs, training (2015 US$)[‡] | | |
| Healthcare personnel (careHPV) | 0.09 per woman screened with careHPV | 50%–150% of base case |
| Laboratory technicians (careHPV) | 0.04 per woman screened with careHPV | 50%–150% of base case |
| Outreach workers/auxiliary nurses (careHPV self-collection) | 0.08 per woman screened with careHPV (self collection) | 50%–150% of base case |
| Healthcare providers (VIA and cryotherapy) | 1.51 per woman receiving VIA and/or cryotherapy | 50%–150% of base case |
| Cost of cancer treatment (2015 US$) (rounded)[29] | | |
| Local cancer | | |
| - Direct medical | 944 | |
| - Direct non-medical** | 197 | |
| - Women's time†† | 346 | |
| Total | 1486 | 944–2229 (direct medical only; 150% of base case) |

**Table 1** Continued

| Test/parameter | Base case | Sensitivity analysis |
|---|---|---|
| Regional and distant cancer | | |
| - Direct medical | 918 | |
| - Direct non-medical** | 390 | |
| - Women's time†† | 640 | |
| Total | 1946 | 918–2920 (direct medical only; 150% of base case) |

Parameter values for sensitivity analysis were determined as follows: screening and triage test performance (cited literature), colposcopy performance (assumption), coverage and compliance (assumptions), treatment eligibility and efficacy (cited literature), discount rate (assumptions), direct medical costs (assumptions), with the exception of colposcopy and cryotherapy, which were based on ranges suggested by Scale-Up and START-UP data), women's costs (assumptions), programmatic costs (assumptions) and cancer costs (assumptions).

*Test performance characteristics of colposcopy in the START-UP demonstration project were derived from the worst diagnosis of the local pathologist relative to the worst diagnosis by a quality control pathologist (gold standard); we applied the treatment threshold of CIN1 , although this was not the treatment threshold in START-UP. To derive test performance of colposcopy, we excluded histological classifications that were inadequate or with a histological classification other than negative, CIN1, CIN2, CIN3 or cancer. Because CIN1 is not a true underlying health state in the microsimulation model, performance of colposcopy in the model is based on the underlying health states of no lesion, HPV infection, CIN2 or CIN3. For a treatment threshold of CIN1, we weighted sensitivity of colposcopy for women with HPV based on the country-specific prevalence of CIN1 among women with HPV infections in the START-UP studies.

†Compliance is defined as the proportion of women who return for each clinical encounter, relative to the previous visit.

‡Unpublished data from the Scale-Up Nicaragua project. Further details on costing data are provided in the online supplementary appendix.

§In sensitivity analysis, we considered the direct medical cost of colposcopy to be equivalent to the cost of colposcopy alone, without biopsy.

¶In sensitivity analysis, we considered the direct medical cost of cryotherapy to include the upper bound of cryotherapy equipment costs (assuming the lowest number of women treated per year per facility, in Carazo, Chontales or Chinandega) and the upper bound of cryotherapy supply costs (assuming the lowest number of women treated per gas tank in any facility in Carazo, Chontales or Chinandega).

**Direct non-medical costs include transportation to a tertiary facility, temporary housing and meals.

††Includes woman's time and support person's time.

ASCUS+, atypical squamous cells of undetermined significance or higher; CIN1+, cervical intraepithelial neoplasia grade 1 or higher; CIN2+, cervical intraepithelial neoplasia grade 2 or higher; HPV, human papillomavirus; HPV+: human papillomavirus test positive; LEEP, loop electrosurgical excision procedure; Pap+: Pap test positive; ST, screen and treat; US$, 2015 US dollars; VIA, visual inspection with acetic acid; VIA+: visual inspection with acetic acid test positive.

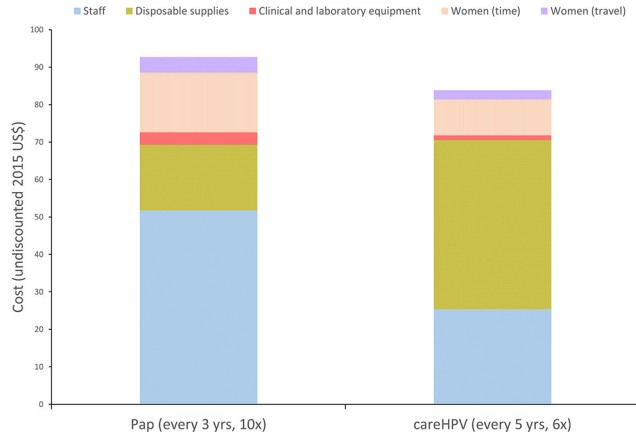

**Figure 2** Cervical cancer screening cost per woman over the duration of screening eligibility, by cost component: Pap testing (every 3 years) versus careHPV testing (every 5 years). Bars indicate the undiscounted cost (2015 US$) of screening with Pap testing (offered 10 times between ages 30 and 59 years) versus careHPV testing (offered six times between ages 30 and 59 years), by cost component. Only screening costs are shown; costs associated with recommended management following a positive screening test are not included. 6×, delivered six times over the course of screening eligible ages 30 to 59; 10×, delivered 10 times over the course of screening eligible ages.

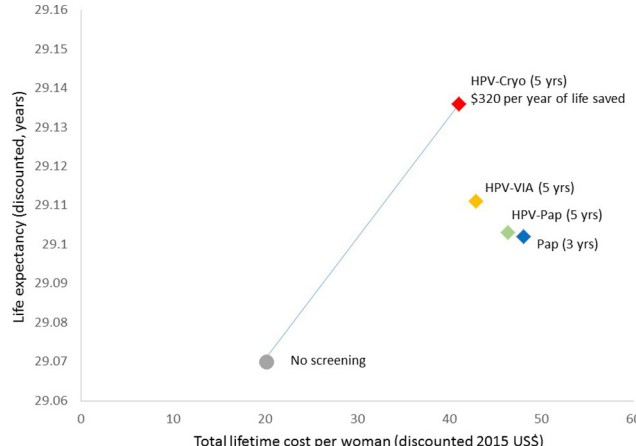

**Figure 3** Cost-effectiveness analysis: base-case results. The graph displays the discounted lifetime costs (x axis; in 2015 US$) and life expectancy (y axis) associated with each screening strategy (Pap testing every 3 years, careHPV every 5 years with cryotherapy for HPV-positive eligible women (HPV-Cryo), careHPV every 5 years with visual inspection with acetic acid (VIA) triage of HPV-positive women (HPV-VIA) and careHPV every 5 years with Pap triage of HPV-positive women (HPV-Pap)), under base-case assumptions. The cost-effectiveness associated with a change from one strategy to a more costly alternative is represented by the difference in cost divided by the difference in life expectancy associated with the two strategies. The curve indicates the strategies that are efficient because they are more effective and either (1) cost less or (2) have a more attractive cost-effectiveness ratio than less effective options. The incremental cost-effectiveness ratio (ICER) is the reciprocal of the slope of the line connecting the two strategies under comparison. In the base-case analysis, HPV-Cryo every 5 years was less costly and more effective than other screening strategies considered and was thus a dominant strategy with an ICER of US$320 per year of life saved. HPV-Cryo, HPV testing with cryotherapy for HPV-positive women; HPV-Pap, HPV testing with Pap triage of HPV-positive women; HPV-VIA, HPV testing with visual inspection with acetic acid triage of HPV-positive women; yrs, years.

risk by 12.2% (range, 10.2%–14.5%). Pap (every 3 years) reduced cancer risk by 10.8% (range, 8.7%–13.4%). Under base-case assumptions, HPV-Cryo was both less costly and more effective than all other strategies, thus dominating HPV-VIA, HPV-Pap and Pap alone. With an ICER of US$320 per year of life saved (YLS), HPV-Cryo every 5 years would be considered 'very cost-effective' given Nicaragua's per capita GDP of US$2090. The total discounted lifetime cost per woman and life expectancy associated with each screening strategy is presented in figure 3.

### Sensitivity analysis

While HPV-Cryo remained the most effective strategy across all sensitivity analyses, the magnitude of reduction in lifetime risk of cancer was dependent on screening coverage of the target population and compliance with recommended follow-up. When coverage was 50% and all other parameters were held constant at base-case values, HPV-Cryo reduced cancer risk by an average of 21.1%; HPV-VIA, HPV-Pap and Pap yielded average cancer risk reductions of 13.9%, 8.7% and 7.7%, respectively. As coverage increased to 80%, all else being equal, HPV-Cryo reduced cancer risk by an average of 33.5%, while HPV-VIA, HPV-Pap and Pap yielded average cancer risk reductions of 22.1%, 14.0% and 12.4%, respectively (see online supplementary appendix). Figure 4 displays the impact of visit compliance on lifetime risk of cancer. When compliance with visits to all facilities (ie, for both screening and referral) was low at 40%, HPV-Cryo remained the most effective strategy but only reduced cancer risk by 16.2%; Pap had little health impact at this

level of compliance, reducing cancer risk by only 5.4%. As compliance at all facilities rose to 85%, HPV-Cryo reduced cancer risk by 47.9%; HPV-VIA, HPV-Pap and Pap reduced cancer risk by 42.0%, 40.7% and 35.7% respectively.

In addition to remaining the most effective strategy across all sensitivity analyses, HPV-Cryo remained the most efficient strategy as well. HPV-Cryo remained the least costly and most effective strategy with a stable ICER of US$320 per YLS when (1) Pap test performance (as a primary screening test) improved, (2) VIA and Pap triage test performance improved, (3) colposcopy was assumed to be perfect and (4) the direct medical cost of LEEP was varied from 75% to 125% of the base case. Despite slight fluctuation in the ICER, HPV-Cryo also remained the least costly and most effective strategy as (1) screening coverage varied from 50% to 80%; (2) visit compliance varied from 40% to 85% per visit; (3) the screen-and-treat cryotherapy cure rate was reduced to

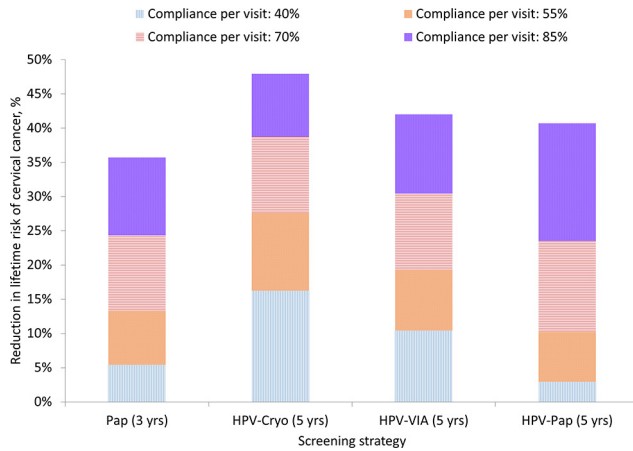

Compliance per visit: 40%  Compliance per visit: 55%
Compliance per visit: 70%  Compliance per visit: 85%

**Figure 4** Reduction in lifetime risk of cervical cancer, by compliance level. Bars indicate the per cent reduction in lifetime risk of cervical cancer for each screening strategy (Pap testing every 3 years, careHPV every 5 years with cryotherapy for HPV-positive women (HPV-Cryo), careHPV every 5 years with visual inspection with acetic acid (VIA) triage of HPV-positive women (HPV-VIA) and careHPV every 5 years with Pap triage of HPV-positive women (HPV-Pap)) as compliance per visit within a screening episode increases. Compliance is defined as the proportion of women who return for each clinical encounter, relative to the previous visit. Coverage of the target population is assumed to be 70%. While the base-case analysis assumed 85% compliance for visits at screening facilities and 40% compliance for visits at referral facilities (for diagnosis and treatment), the graph displays cancer risk reduction assuming the specified compliance level at all visits, regardless of facility type.

75%; (4) eligibility for cryotherapy was reduced to 75% or less; (5) the treatment cure rate following colposcopy was reduced to 85%; (6) women receiving treatment following colposcopy in the Pap and HPV-Pap strategies were assumed to receive cryotherapy instead of the more costly LEEP; (7) the discount rate was varied from 0% to 5%; (8) only payer costs were considered, excluding women's time and transportation costs; (9) the direct medical cost of HPV self-collection was varied from 75% to 125% of the base case; (10) the direct medical cost of cryotherapy was increased to 170% of the base case; (11) the direct medical cost of colposcopy was reduced to 35% of the base case; (12) programmatic costs associated with HPV-based screening were varied from 50% to 150% of the base case; (13) women's time and transportation costs were reduced to 50% of the base case; and (14) the costs of cancer treatment ranged from including only direct medical costs to 150% of the base case (figure 5). Among variables considered, compliance per visit appears to have the greatest impact on the ICER for HPV-Cryo, with 40% compliance yielding an ICER of US$580 per YLS and 85% compliance yielding an ICER of US$190 per YLS. Even when visit compliance is low, HPV-Cryo would be considered very cost-effective.

When only payer costs were considered (ie, women's time and transportation costs were excluded), the total lifetime cost per woman was lower for all strategies and Pap

every 3 years was slightly less costly (although still slightly less effective) than HPV-Pap every 5 years. HPV-Cryo remained the most effective and efficient strategy, with an ICER of US$270 per YLS (see online supplementary appendix).

The only scenario in which Pap testing every 3 years was the least costly strategy occurred when the direct medical cost of Pap testing was US$3 (base case, US$7.26), a value commonly cited for the cost of Pap in Nicaragua, though the source of this estimate is unknown. However, Pap remained the least effective strategy, and HPV-Cryo had a lower cost-effectiveness ratio, maintaining an ICER of US$320 per YLS (see online supplementary appendix).

### Scenario analysis: HPV-Cryo unavailable

When we assumed HPV-Cryo was not available as a screening strategy, HPV-VIA was the least costly and most effective strategy in the base-case and most sensitivity analyses, with a base-case ICER of US$550 per YLS (see online supplementary appendix). Exceptions included the following circumstances: (1) when the direct medical cost of Pap was US$3, Pap alone had a more attractive ICER (US$530), although was less effective than HPV-VIA (US$630 per YLS); (2) Pap performance in the general screening population was improved and Pap alone became the most effective strategy, with an ICER of US$540 per YLS; (3) VIA test sensitivity in HPV-positive women was only 0.40, in which case HPV-VIA had an ICER of US$726 per YLS, but HPV-Pap was more effective with an ICER of US$3260.

### DISCUSSION

Using implementation data from the Scale-Up project—which aims to facilitate institutionalisation of HPV testing at the national level in Guatemala, Honduras and Nicaragua—we estimated the long-term health impact and value of careHPV testing in Nicaragua's public health system. We found that screening algorithms consisting of HPV testing at 5-year intervals would be less costly and more effective than screening with Pap testing at 3-year intervals. Furthermore, HPV testing followed by treatment with cryotherapy for all eligible HPV-positive women would be less costly and more effective than HPV testing followed by triage testing with either VIA or Pap for HPV-positive women. A screen-and-treat HPV programme would be a very cost-effective intervention in Nicaragua, with an ICER of US$320 per YLS under base-case assumptions. These findings were robust across sensitivity analyses. The comparatively large health benefits and efficiency of HPV-Cryo can largely be attributed to the relatively low number of visits to healthcare facilities and the high sensitivity of the careHPV test to detect both CIN2 and oncogenic HPV infections with the potential to develop into precancer.

We found that screening coverage of the target population had a considerable impact on achievable reductions in cervical cancer risk, with HPV-Cryo yielding the

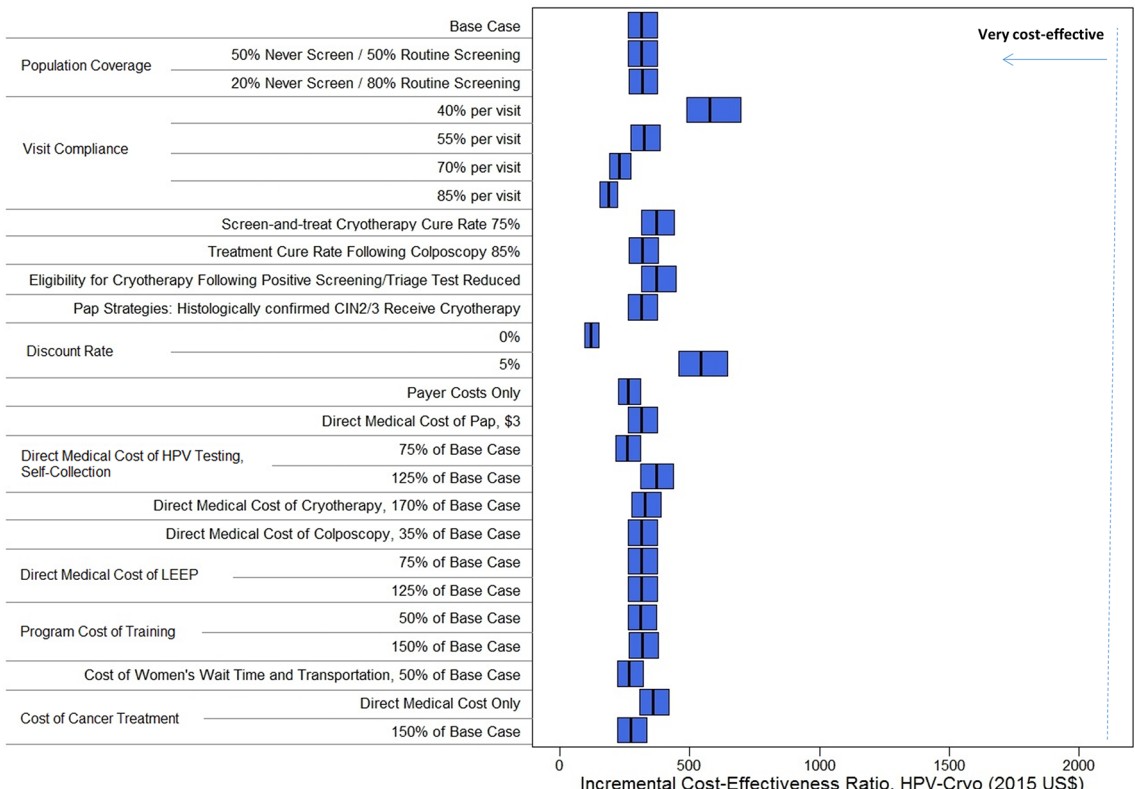

**Figure 5** Base-case and sensitivity analyses: incremental cost-effectiveness ratios, HPV cryotherapy (HPV-Cryo) strategy. Incremental cost-effectiveness ratios (ICER) are presented (x-axis, 2015 US$ per year of life saved) for the base-case and sensitivity analyses (y-axis). The blue bars represent the range of the ICER for HPV-Cryo every 5 years across the 50 input parameter sets, with the ICER of the mean costs divided by the mean effects demarcated by a black line. The dashed blue line indicates Nicaragua's per capita gross domestic product (GDP), at US$2,090, assuming this is the threshold that designates interventions as 'very cost-effective'.

greatest risk reduction. Due to proportional increases in both costs and health benefits, the ICER for HPV-Cryo remained stable as coverage increased from 50% to 80%. Compliance with recommended follow-up was a key driver of both achievable reductions in cancer risk and the ICER of HPV-Cryo. As the proportion of women who returned for each clinical encounter (relative to the previous visit) increased from 40% to 85%, the effectiveness or cancer benefit associated with HPV-Cryo rose from 16.2% to 47.9% as more women were linked to treatment; the ICER fell from US$580 per YLS to US$190 per YLS as more cancers were averted. Thus, improved efforts to successfully navigate women to recommended follow-up will enhance screening programme effectiveness and efficiency.

While substantially reducing the cost of Pap testing to US$3 (less than half of the base case value) made Pap the strategy with the lowest per-woman lifetime costs, the confluence of low test sensitivity and the high number of health facility visits needed to complete screening, diagnostic follow-up and treatment made Pap the least effective strategy since women are lost to follow-up with each additional required visit. Even in this low-cost Pap scenario, HPV-Cryo remained the most- effective and cost-effective strategy.

When we assumed a scenario in which HPV-Cryo was not available for logistic and programming reasons, we found that HPV-VIA was usually the least costly and most effective of the remaining strategies. Although the ICER was less attractive than the ICER associated with HPV-Cryo in the main analysis, it was below Nicaragua's per capita GDP.

There are several limitations to this analysis. We did not consider alternative screening intervals or ages for each strategy, but rather restricted the analysis to the ages and intervals currently under consideration by the Nicaraguan Ministry of Health. It is likely that an increased screening interval and fewer lifetime screens will also be cost-effective, although health benefits may be reduced; we demonstrated in a previous analysis that screening once or three times in a woman's lifetime with careHPV would be very cost-effective in Nicaragua.[17] The screening algorithms, as modelled, reflect the prototypical structure of a screening episode and the type of facility at which visits usually take place, but do not capture variation due to geography or health facility capacity. Furthermore, the modelled screening algorithms do not entirely reflect the complex downstream follow-up of screen-positive women that is embodied in the Ministry of Health's screening guidelines. In

simplifying the downstream follow-up for modelled strategies, we may have underestimated the costs and overestimated the benefits relative to the national guidelines, which call for additional follow-up prior to treatment. However, the modelled strategies would likely bias the analysis in favour of Pap and HPV triage strategies. We also did not consider a strategy in which women with ASCUS were referred directly to cryotherapy instead of colposcopy because that approach is not recommended by the WHO guidelines or by any professional medical society to the best of our knowledge. We did not consider quality of life impact nor the potential disutilities or harms that might be associated with overtreatment.

While we adopted a micro-costing approach to leverage data from the START-UP and Scale-Up projects in Nicaragua, there remain limitations to our cost estimates. First, individual-level data for each woman were not available; thus, our estimates represent average costs in the project populations. Second, we did not have information on the costs associated with HPV self-collection in community settings, where most self-collection takes place. Instead, we assumed all self-collection took place at the clinic. Compared with clinic-based efforts, community-based self-collection may be associated with lower costs for women's time and travel, and higher direct medical and programmatic costs due to outreach worker involvement in facilitating screening and delivering results. Third, our estimates of programmatic costs were restricted to training sessions, and we did not have information on the costs of social mobilisation and outreach, patient navigation and support or infrastructural improvements that would be required to successfully scale up a screening programme. Fourth, we valued women's time based on the minimum wage in Nicaragua. This may be a conservative estimate if most women attending screening are formally employed; conversely, it may overestimate the societal value of women's time spent working in the informal sector or at home. Finally, we extrapolated the cost of cancer treatment using data from El Salvador.[29] Despite these limitations, extensive sensitivity analyses on cost components indicate that HPV-Cryo is robustly the most efficient strategy.

As implementation of HPV testing continues, particularly without triage testing, the health system's capacity to provide cryotherapy will likely need to increase. While the use of triage testing (either with Pap or HPV) reduces the number of cryotherapy procedures performed, we found that the lower sensitivity of triage testing (resulting in more false negatives) led to a decline in health benefits as fewer women with persistent HPV infection and precancer received treatment. The cost savings associated with fewer cryotherapy procedures were outweighed by increased costs of additional follow-up and cancer treatment in triage-negative women. A sensitivity analysis on the cost of cryotherapy revealed that HPV-Cryo remained the dominant strategy even when costs increased to 170% of the base case. However, we did not explicitly consider the costs of increasing access to cryotherapy machines or the implications of gas stock-outs, which have been identified as barriers in some low-income and middle-income countries.[40] New ablative technologies currently undergoing testing are smaller, portable and do not require gas. Thermal-coagulation has been used in the UK for more than 30 years, and now it is being used in several low-income and middle-income countries, including as part of a 'screen-and-treat' programme in Malawi,[41] and is currently undergoing testing in Latin America. If newer technologies demonstrate cure rates similar to cryotherapy, the cost-effectiveness of screen-and-treat algorithms may improve along with access to treatment.

In summary, using data from the Scale-Up implementation project in Nicaragua, we found that HPV testing followed by cryotherapy for eligible HPV-positive women (a screen-and-treat approach) was a very cost-effective intervention in Nicaragua. As the HPV-Cryo algorithm was not implemented in phase 2 of the Scale-Up project, compliance and cost estimates may need to be further honed to reflect improvements in capacity for cryotherapy if HPV-Cryo is implemented going forward. While it is too early to assume that costs and health impact from phase 2 of implementation are generalisable to other departments in Nicaragua or other settings in Central America, extensive sensitivity analyses indicate the robustness of findings. An HPV-based screening algorithm involving a similar screen-and-treat approach was recently found to be a good value for public health dollars in El Salvador,[29] where a national scale up is underway. It is important to note that a favourable cost-effectiveness profile does not guarantee that HPV-Cryo will be affordable or feasible in a lower-middle-income country like Nicaragua. Both the cost-effectiveness ratio, budgetary impact and health system infrastructure need to be favourable for screening programs to be sustainable. We present these findings to inform evidence-based decision making around national screening guidelines, programme design and implementation and budgeting for infrastructural improvements and procurement of HPV tests in Nicaragua.

**Acknowledgements** We gratefully acknowledge the efforts of Movicáncer of Nicaragua in gathering costing data.

**Contributors** All authors developed the analysis plan and interpreted the data. NGC, MM, JJ, FH and JJK conceptualised the study. MM, JJ, FH and EV collected implementation data. NGC conducted data analysis and wrote the first draft of the report and revised subsequent drafts. JJ was the principal investigator of the START-UP and Scale-Up projects. JJK was the principal investigator overseeing microsimulation model development. All authors contributed to and approved the final report.

**Funding** This work was supported by the Bill & Melinda Gates Foundation.

**Disclaimer** The findings and conclusions contained within are those of the authors and do not necessarily reflect positions or policies of the Bill & Melinda Gates Foundation. The funders had no role in study design; data collection, analysis and interpretation; preparation of the manuscript; or decision to submit the article for publication.

**Competing interests**  JJ was the director of the START-UP demonstration projects and received all tests used in the study as a donation from Qiagen; no other relationships or activities that could appear to have influenced the submitted work. JJ was the co-owner and Deputy Manager of Onco Prev International, a Peruvian company, from 2012 through March 2017. Onco Prev International offers cervical cancer screening services and in 2016 also began positioning for distribution of medical devices including colposcopes and the Liger thermocoagulator. Onco Prev International did not commercialize any medical instrument during the time JJ was part of the company.

**Provenance and peer review**  Not commissioned; externally peer reviewed.

**Data sharing statement**  PATH provides technical assistance to the government of Nicaragua and has access to screening indicators through partnership with Movicáncer, a local non-governmental organisation partner in Nicaragua. Movicáncer designed the health information system used by the Ministry of Health to track screening and treatment of women's cancers. The Ministry of Health protects individually identifiable information; PATH received de-identified and consolidated data on visit compliance from Movicáncer. Costing data were collected by Movicáncer through consultation with Ministry of Health personnel, mostly via phone interviews. No individual identifiers were collected. These data are not available from any public source. A supplementary appendix has been provided to describe costing data and the microsimulation model in detail.

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
