## [Reviewer comments · BMJ Open]

ARTICLE DETAILS

TITLE (PROVISIONAL)	The Cost-Effectiveness of HPV-Based Cervical Cancer Screening in the Public Health System in Nicaragua
AUTHORS	Campos, Nicole; Mvundura, Mercy; Jeronimo, Jose; Holme, Francesca; Vodicka, Elisabeth; Kim, Jane

VERSION 1 - REVIEW

REVIEWER	Talía Malagón McGill University, Canada
REVIEW RETURNED	13-Dec-2016

GENERAL COMMENTS	In this paper Campos et al. present us a cost-effectiveness analysis of cervical cancer screening strategies in Nicaragua. Improving cervical cancer screening in Nicaragua is clearly a worthy public health target given that cervical cancer is the most incident female cancer in Nicaragua, is the leading cause of female cancer mortality, and there is no current HPV vaccination program in Nicaragua. Overall, the analysis is well carried out and provides conclusions that are robust to various sensitivity analyses the authors have performed. The authors have paid close attention to the pathways of care required by different screening strategies and the costs involved with each. It is particularly valuable to have used a societal perspective in the base case of their analysis, as this perspective is not always considered in cost-effectiveness analyses of cervical cancer prevention. The analysis is clearly presented in both the main text and the appendix. I only have a few comments regarding some points that were less clear or that warrant further discussion: • The effectiveness and ICERs of cervical cancer screening are substantially influenced by the compliance with recommended follow-up, regardless of chosen screening test and triage strategy (Figures 4-5). This warrants some further highlighting and discussion, as screening is unlikely to substantially reduce cancer incidence unless there is high compliance. Why does cost-effectiveness increase with compliance? Is it because cases lost to follow-up lead to more wasted resources? Also, why is compliance per visit so low? Have the scale up projects underway and national guidelines considered the issue of how to increase compliance as well as the methods and algorithms for screening?• The authors should briefly discuss the generalizability of results to other countries and/or the other departments of Nicaragua.• Table 1: Where is the line for the test performance of self-collected HPV samples?• Figure 2: Why are only the screening costs shown in this figure? We need to have the total undiscounted costs of all strategies predicted by the model (including management costs), as recommended by CHEERS. Especially as one of the aims of the paper is to estimate the economic cost of cervical cancer screening
--

	with careHPV testing, which will include management costs. Else, the authors should explain why they are only presenting screening costs.  • Figures 2, 4, & A3: These figures do not print well in black and white. The authors may want to consider a more graded color scheme or adding patterns that will still make the figures understandable in black & white.
--	---

REVIEWER	Gaby Sroczynski Institute of Public Health, Medical Decision Making and Health Technology Assessment, UMIT - University for Health Sciences, Medical Informatics and Technology, Austria Dr. Sroczynski reports grants from COMET Center ONCOTYROL, Austria, funded by the Austrian Federal Ministries BMVIT/BMWFJ (via FFG) and the Tiroler Zukunftsstiftung/Standortagentur Tirol (SAT), grants from the German Agency for HTA at the German Institute for Medical Documentation and Information (DAHTA@DIMDI), an Institute of the German Federal Ministry of Health in Germany, grants from the Institute for Quality and Efficiency in Health Care (IQWiG) in Germany, grants from H2020 EU, and expert advise for S3 Guidelines Committee, Germany, and the Health Information and Quality Authority (HIQA) in Ireland.
REVIEW RETURNED	16-Dec-2016

GENERAL COMMENTS	The authors performed a decision analysis using Monte Carlo simulation in order to evaluate longterm effectiveness and cost-effectiveness of different cervical cancer screening strategies within the health care context of Nicaragua. Specifically, the author's objectives were 1) estimate the economic cost of cervical cancer screening with careHPV testing, and 2) project the long-term health and economic impact and value of careHPV testing in Nicaragua relative to existing Pap-based screening. The research topic is important for guidance of health care services in Nicaragua. The methods used are suitable and performed according to international guidelines for modelling studies. The model was calibrated and validated against observed data. The most important key outcome of this study is that numbers of follow-up visits are associated with decreased effectiveness (due to a combination of low compliance and lower detection/treatment rates) and increased costs (due to increased follow-up costs and treatment costs for LEEP). Therefore, a straight forward screen and treat algorithm provides the most efficient screening option. In the following sections the main aspects that may need revision before publication are described. Minor Revisions  • Input data/model structure: a prior publication was cited for details in the model structure and input parameter for natural history. However, it would be nice for the reader to have at least a table with natural history data and a state-transition diagram for the model structure in the Appendix. • Input data: the likelihood of ASCUS/LSIL/HSIL in CIN1/CIN2/CIN3/Cancer are not reported • Input data: likelihood of a positive primary screening test in CIN1? • Input data: ranges for sensitivity analyses are reported, but it is not
--

	stated whether those are assumptions or abstracted from the standard variation of base case data used  • Input data/strategies: eligibility for cryotherapy in CIN2 or 3: are these assumptions? Why are these not varied in sensitivity analyses? In the strategies HPV-Pap and Pap women with colposcopy diagnosed CIN2/3 are all treated with LEEP (which is more costly). What is the reason for not considering a proportion being eligible for cryotherapy? • Strategies: direct referral to cryotherapy after ASCUS+ or HPV+/ASCUS+ are not considered. Colposcopy directed treatment process is associated with increased numbers of necessary FU visits (associated with low compliance) and therefore lower effectiveness and in addition with increased costs (costs for colposcopy are even slightly higher than for cryotherapy and proportion of women treated with relatively costly LEEP is high). Authors may further discuss. • Strategies: different screening intervals or different screening starting age are not evaluated. In the discussion section it was reported as a limit. Authors may also discuss that strategies with increased screening interval or once in a lifetime at a specific women's age may also derive acceptable balance between benefits and costs. • Analyses: analyses techniques and software program used are not reported in the analysis section of the methods • Analysis: LEEP cost were not varied in sensitivity analyses • Analyses: discount rate not varied in sensitivity analyses • Analyses: screening and treatment related disutilities and disease-related quality-of-life reduction not considered. It should be discussed that the analyses did not thoroughly consider benefit-harm relation. This is especially important, as the favored treatment is cryotherapy, which may have negative side effects. Table 1  • Total cost of local cancer: 1,486 -> sum: 1,487; rounding error? Please state "rounded" or report cents • Total cost of local cancer: range 2,2920 -> should be 2,229 (150% of base case costs) • Total cost of regional/distant cancer: 1,946 -> sum: 1,948; rounding error? Please state "rounded" or report cents • Total cost of regional/distant cancer: range 2,290 -> should be 2,919 (150% of base case costs) Figure 1  • Legend: ASCUS description is missing Figure 3  • Title: Cost-effectiveness analysis: Base case results Figure 5  • Why are there no bars shown for population coverage, visit compliance, direct medical cost of HPV testing self-collection, programmatic cost of training, costs of cancer treatment? Are those ICERs above the per capita GDP of Nicaragua? How much? References  • Ref 17: pages etc? • Ref 24: URL?
--	--

REVIEWER	Iacopo Baussano International Agency for Research on Cancer World Health Organization 150 cours Albert Thomas 69372 Lyon CEDEX 08, France
REVIEW RETURNED	04-Jan-2017

GENERAL COMMENTS	The manuscript “The Cost-Effectiveness of HPV-Based Cervical Cancer Screening in the Public Health System in Nicaragua” by Nicole Campos and colleagues is excellent. The study methodology is solid, the presentation is transparent, and the number of screening alternatives compared is reasonable. The use of implementation data from the Scale-Up project makes the study findings convincing and up-to-date. Major comments I have only one major comment for the authors. Although the specific context of the study is the Cervical Cancer Screening program in Nicaragua, the findings reported in the manuscript are likely to be of interest also in other resource-limited settings. I expect that in several resource-limited settings HPV-ST strategy will not be implemented for a range of logistic and programmatic reasons. From this perspective, it would be useful to have more detailed analyses, in particular sensitivity analyses (such as in figure 5), and a direct comparison of HPV-VIA and HPV-Pap screening strategies. According to figures 3, A4, and A5, HPV-VIA dominates HPV-Pap. It is however crucial to assess the sensitivity of this finding across a range of assumptions, in particular about triage test performances. Both Pap and VIA have shown a wide range of performance variability. Furthermore, VIA is likely to become less sensitive with age as the squamo-columnar junction recedes into the endocervical canal. A direct comparison of HPV-VIA and HPV-Pap is likely to broaden this interest and usefulness of the paper. Incidentally, I am not sure how to interpret the values reported in table 1 (see below). Minor Comments - The definition “HPV screen-and-treat” is somehow misleading, as it suggest that HPV detection and cryotherapy for HPV-positive occurs within the same visit. “HPV-Cryo” would be less confusing. - Table 1, column “sensitivity analysis”. It is not clear which sensitivity and specificity ranges were explored. To avoid confusion I suggest to use the following notation “x to y” or “x - y” to denote a range. - Page 16, lines 328-29 “individual-level data for each woman were not available; thus, our estimates represent average costs in the project populations”. Does this mean that ONLY average costs were available (as I expect) or that you had access to a distribution of the costs. In the latter case, you could in principle account for such distribution in your analysis. - Figure 5 may be improved: a) Groups of variables are difficult to identify (subheadings should be better identifiable); b) a large fraction the plot area (from 750 USD to 2000 USD on the horizontal axis) is almost non-informative. - Refs 28 and 42 refer to the same paper, i.e. Ronco et al, Lancet 2014.
---

VERSION 1 – AUTHOR RESPONSE

Reviewer 1: Talía Malagón

1. In this paper Campos et al. present us a cost-effectiveness analysis of cervical cancer screening strategies in Nicaragua. Improving cervical cancer screening in Nicaragua is clearly a worthy public

health target given that cervical cancer is the most incident female cancer in Nicaragua, is the leading cause of female cancer mortality, and there is no current HPV vaccination program in Nicaragua. Overall, the analysis is well carried out and provides conclusions that are robust to various sensitivity analyses the authors have performed. The authors have paid close attention to the pathways of care required by different screening strategies and the costs involved with each. It is particularly valuable to have used a societal perspective in the base case of their analysis, as this perspective is not always considered in cost-effectiveness analyses of cervical cancer prevention. The analysis is clearly presented in both the main text and the appendix.

We thank the reviewer for his/her positive feedback.

2. The effectiveness and ICERs of cervical cancer screening are substantially influenced by the compliance with recommended follow-up, regardless of chosen screening test and triage strategy (Figures 4-5). This warrants some further highlighting and discussion, as screening is unlikely to substantially reduce cancer incidence unless there is high compliance. Why does cost-effectiveness increase with compliance? Is it because cases lost to follow-up lead to more wasted resources? Also, why is compliance per visit so low? Have the scale up projects underway and national guidelines considered the issue of how to increase compliance as well as the methods and algorithms for screening?

The importance of compliance and reasons for efficiency gains as compliance rises are highlighted prominently in the Discussion, and we have added some text to clarify:

p. 16: "Compliance with recommended follow-up was a key driver of both achievable reductions in cancer risk and the ICER of HPV-Cryo. As the proportion of women who returned for each clinical encounter (relative to the previous visit) increased from 40% to 85%, the effectiveness or cancer benefit associated with HPV-ST rose from 16.2% to 47.9% as more women were linked to treatment; the ICER fell from US\$580 per YLS to US\$190 per YLS as more cancers were averted. Thus, improved efforts to successfully navigate women to recommended follow-up will enhance screening program effectiveness and efficiency."

As mentioned above, the ICER for HPV-Cryo (formerly called HPV-ST) decreases (i.e., becomes more attractive) when compliance improves primarily because effectiveness (i.e., the cancer benefit) increases dramatically as more women are linked to treatment. As compliance varies between 40% and 85% per visit, total lifetime cost per woman fluctuates slightly as costs associated with follow-up visits and procedures are weighed against cost savings from averted cancers, but effectiveness is the primary driver that impacts cost-effectiveness.

The compliance per visit was relatively low in the Scale-Up project because it was not a research study, but rather, an implementation project within the public health care system in Nicaragua. The need for screen-positive women to receive treatment at a referral facility, rather than their usual health care facility closer to home, posed a significant burden on the average woman due to time and transportation costs. Improving follow-up is a priority as implementation continues, and different options are being explored, including the use of cell phones to help navigate women to care.

3. The authors should briefly discuss the generalizability of results to other countries and/or the other departments of Nicaragua.

We have added the following text:

p. 19: “As the HPV-Cryo algorithm was not implemented in Phase 2 of the Scale-Up project, compliance and cost estimates may need to be further honed to reflect improvements in capacity for cryotherapy if HPV-Cryo is implemented going forward. While it is too early to assume that costs and health impact from Phase 2 of implementation are generalizable to other departments in Nicaragua or other settings in Central America, extensive sensitivity analyses indicate the robustness of findings. An HPV-based screening algorithm involving a similar “screen-and-treat” approach was recently found to be good value for public health dollars in El Salvador [29], where national scale-up is underway.”

4. Table 1: Where is the line for the test performance of self-collected HPV samples?

We have corrected the typo in Table 1 to reflect the test performance of self-collected vaginal HPV samples (this previously read “provider-collected vaginal HPV samples”).

5. Figure 2: Why are only the screening costs shown in this figure? We need to have the total undiscounted costs of all strategies predicted by the model (including management costs), as recommended by CHEERS. Especially as one of the aims of the paper is to estimate the economic cost of cervical cancer screening with careHPV testing, which will include management costs. Else, the authors should explain why they are only presenting screening costs.

Figure 2 (“Cervical cancer screening cost per woman over duration of screening eligibility, by cost component: Pap testing (every 3 years) versus careHPV testing (every 5 years)”) is included as part of the Methods section on cost data, and accordingly represents the estimated total screening cost per woman over her lifetime (i.e., for screening visits only, to show the difference in model cost inputs), depending on whether the screening strategy is Pap or HPV. This figure is not intended to present the total lifetime cost per woman that is generated by the model, as these are presented in Figure 3. Rather, our objective in displaying Figure 2 is to show the relative breakdown of screening costs, which are a key driver of the total lifetime cost per woman. As shown, the need for more frequent screening with Pap testing and the greater burden on women’s costs results in greater screening costs per woman.

6. Figures 2, 4, & A3: These figures do not print well in black and white. The authors may want to consider a more graded color scheme or adding patterns that will still make the figures understandable in black & white.

We have modified the color scheme and added patterns to ensure that segments of the bar graphs can be distinguished when printed in black and white.

Reviewer 2: Gaby Sroczyński

1. Input data/model structure: a prior publication was cited for details in the model structure and input parameter for natural history. However, it would be nice for the reader to have at least a table with natural history data and a state-transition diagram for the model structure in the Appendix.

While the state-transition diagram was presented in the cited prior publication (Campos et al. 2014) and we would need to acquire permission to present it here, we have added the comprehensive table of transition probabilities in the natural history model to the Appendix (Table A.3).

2. Input data: the likelihood of ASCUS/LSIL/HSIL in CIN1/CIN2/CIN3/Cancer are not reported

The Pap screening strategy does not have this level of granularity because the management algorithm did not depend on Pap result if a woman had ASCUS or higher. We assumed that all women with ASCUS and higher would be referred to colposcopy (described on Page 10 of the manuscript), and the test performance of Pap was based on the probability of ASCUS or higher given a true health state of cervical intraepithelial neoplasia grade 2 or higher (CIN2+), as reported in the literature.

3. Input data: likelihood of a positive primary screening test in CIN1?

CIN1 is not a health state in our natural history model. Cervical cancer experts have agreed that CIN1 is a microscopic manifestation of acute HPV infection and is therefore incorporated into the HPV-infected state. Recent literature on test performance has focused on the probability of a positive test result given the presence of CIN2+, and few recent test performance studies have reported the probability of testing positive for women with CIN1. For Pap and VIA, test performance in our model depends on the presence or absence of CIN2+. For HPV testing, test performance depends on the presence or absence of oncogenic HPV, and we calibrate the probability of testing positive or negative for oncogenic HPV given the prevalence of oncogenic HPV in women with no lesion versus women with CIN2+.

4. Input data: ranges for sensitivity analyses are reported, but it is not stated whether those are assumptions or abstracted from the standard variation of base case data used.

Ranges explored in sensitivity analysis are based on informed assumptions where data are lacking (e.g., visit compliance, for which we explored a wide range of proportions; selected costing data where we varied costs by some percentage of the base case), or, in other cases, the reasonable variation suggested by the published literature, for which the citations are included (e.g., screening and diagnostic test performance; eligibility for cryotherapy). For the costs of cryotherapy and colposcopy/biopsy, we explored the implications of specific values if, for instance, none of colposcopies were accompanied by a biopsy or if we used the upper bound cryotherapy costs suggested by the data (as noted in Table 1 footnotes "e" and "f").

We have added text to footnote "a":

Table 1, fn "a": "Parameter values for sensitivity analysis were determined as follows: screening and triage test performance (cited literature); colposcopy performance (assumption); coverage and compliance (assumptions); treatment eligibility and efficacy (cited literature); discount rate (assumptions); direct medical costs (assumptions, with the exception of colposcopy and cryotherapy, which were based on ranges suggested by Scale-Up and START-UP data); women's costs (assumptions); programmatic costs (assumptions); cancer costs (assumptions)."

5. Input data/strategies: eligibility for cryotherapy in CIN2 or 3: are these assumptions? Why are these not varied in sensitivity analyses? In the strategies HPV-Pap and Pap women with colposcopy diagnosed CIN2/3 are all treated with LEEP (which is more costly). What is the reason for not considering a proportion being eligible for cryotherapy?

We have added a sensitivity analysis on eligibility for cryotherapy in which we assume a much lower

proportion of women are eligible for cryotherapy; the reference for this alternative range has been added to Table 1 (Gage et al., 2009). We found that, while the ICER for HPV-Cryo increased slightly, it remained a dominant strategy. We have added the findings from this sensitivity analysis to the Results section and Figure 5.

In the Pap and HPV-Pap strategies, we have added a sensitivity analysis exploring the extreme scenario in which all women receiving treatment after colposcopy receive cryotherapy, but with the effectiveness of LEEP. This should bias the analysis in favor of Pap-related screening. We found that this did not change the greater attractiveness of HPV-Cryo. We have added the findings from this sensitivity analysis to the Results section and Figure 5.

6. Strategies: direct referral to cryotherapy after ASCUS+ or HPV+/ASCUS+ are not considered. Colposcopy directed treatment process is associated with increased numbers of necessary FU visits (associated with low compliance) and therefore lower effectiveness and in addition with increased costs (costs for colposcopy are even slightly higher than for cryotherapy and proportion of women treated with relatively costly LEEP is high). Authors may further discuss.

In sensitivity analyses, we explore the impact of 1) reducing the cost of colposcopy to 35% of the base case; 2) increasing follow-up to 85% per visit; and 3) assuming that women referred to treatment after colposcopy in the Pap and HPV-Pap strategies receive a treatment with the cost of cryotherapy but the efficacy of LEEP (new sensitivity analysis). HPV-Cryo remains the least costly and most effective strategy in each of these scenarios. Not only are the Pap-related strategies hampered by visit compliance and costly follow-up procedures, but sensitivity of Pap is low, and fewer women with potentially carcinogenic HPV infections are treated.

We have added the following text to the Discussion of limitations to address this concern:

p. 17: "We also did not consider a strategy in which women with ASCUS+ were referred directly to cryotherapy instead of colposcopy because that approach is not recommended by the WHO guidelines, or by any professional medical society to the best of our knowledge."

7. Strategies: different screening intervals or different screening starting age are not evaluated. In the discussion section it was reported as a limit. Authors may also discuss that strategies with increased screening interval or once in a lifetime at a specific women's age may also derive acceptable balance between benefits and costs.

We have added the following text to the Discussion to address this issue:

p. 17: "We did not consider alternative screening intervals or ages for each strategy, but rather restricted the analysis to the ages and intervals currently under consideration by the Nicaraguan Ministry of Health. It is likely that an increased screening interval and fewer lifetime screens will also be cost-effective, although health benefits may be reduced; we demonstrated in a previous analysis that screening once or three times in a woman's lifetime with careHPV would be very cost-effective in Nicaragua [17]."

8. Analyses: analyses techniques and software program used are not reported in the analysis section of the methods

We describe the analytic methods for incremental cost-effectiveness analysis on p. 8. We have added

the following text to the Methods to report the software program used:

p. 8: "These cost data were input into a previously developed Monte Carlo simulation model (programmed in C++) of the natural history of HPV infection and cervical cancer that was calibrated to epidemiologic data from Nicaragua [17, 18]. We then used the model to project the lifetime health and economic outcomes associated with careHPV testing, using three different algorithms for the management of women who test HPV-positive, and Pap-based screening for women aged 30 to 59 years."

9. Analysis: LEEP cost were not varied in sensitivity analyses

We have added a sensitivity analysis in which the cost of LEEP is varied from 75% to 125% of the base case. Base case findings did not change. We have added this new sensitivity analysis to the Results section, Table 1, and Figure 5.

10. Analyses: discount rate not varied in sensitivity analyses

We have added a sensitivity analysis in which the discount rate is varied from 0% to 5%. We have added this new sensitivity analysis to the Results section, Table 1, and Figure 5.

11. Analyses: screening and treatment related disutilities and disease-related quality-of-life reduction not considered. It should be discussed that the analyses did not thoroughly consider benefit-harm relation. This is especially important, as the favored treatment is cryotherapy, which may have negative side effects.

We have added the following text to the limitations mentioned in the Discussion:

p. 17: "We did not consider quality of life impact or the potential disutilities or harms that might be associated with overtreatment."

12. Table 1: Total cost of local cancer: 1,486 -> sum: 1,487; rounding error? Please state "rounded" or report cents

We have noted in the Table that cancer costs have been rounded.

13. Table 1: Total cost of local cancer: range 2,2920 -> should be 2,229 (150% of base case costs)

We have corrected the error.

14. Table 1: Total cost of regional/distant cancer: 1,946 -> sum: 1,948; rounding error? Please state "rounded" or report cents

We have noted in the Table that cancer costs have been rounded.

15. Table 1: Total cost of regional/distant cancer: range 2,290 -> should be 2,919 (150% of base case costs)

We have corrected the error.

16. Figure 1: Legend: ASCUS description is missing

We have added the ASCUS description to the figure.

17. Figure 3: Title: Cost-effectiveness analysis: Base case results

We have made the suggested change.

18. Figure 5: Why are there no bars shown for population coverage, visit compliance, direct medical cost of HPV testing self-collection, programmatic cost of training, costs of cancer treatment? Are those ICERs above the per capita GDP of Nicaragua? How much?

There are bars for different ranges of these variables; bars are presented for each value considered below the heading. We have clarified the figure so that all values considered under a particular heading are more clearly grouped.

19. References: Ref 17: pages etc? Ref 24: URL?

We have added the missing information to these two references.

Reviewer 3: Iacopo Baussano

1. The manuscript "The Cost-Effectiveness of HPV-Based Cervical Cancer Screening in the Public Health System in Nicaragua" by Nicole Campos and colleagues is excellent. The study methodology is solid, the presentation is transparent, and the number of screening alternatives compared is reasonable. The use of implementation data from the Scale-Up project makes the study findings convincing and up-to-date.

We thank the reviewer for the positive feedback.

2. I have only one major comment for the authors. Although the specific context of the study is the Cervical Cancer Screening program in Nicaragua, the findings reported in the manuscript are likely to be of interest also in other resource-limited settings. I expect that in several resource-limited settings HPV-ST strategy will not be implemented for a range of logistic and programmatic reasons. From this perspective, it would be useful to have more detailed analyses, in particular sensitivity analyses (such as in figure 5), and a direct comparison of HPV-VIA and HPV-Pap screening strategies. According to figures 3, A4, and A5, HPV-VIA dominates HPV-Pap. It is however crucial to assess the sensitivity of this finding across a range of assumptions, in particular about triage test performances. Both Pap and VIA have shown a wide range of performance variability. Furthermore, VIA is likely to become less sensitive with age as the squamo-columnar junction recedes into the endocervical canal. A direct comparison of HPV-VIA and HPV-Pap is likely to broaden this interest and usefulness of the paper. Incidentally, I am not sure how to interpret the values reported in Table 1 (see below).

We have added a “scenario analysis” to the manuscript, in which we evaluate the scenario that HPV-Cryo is not available, but Pap, HPV-VIA, and HPV-Pap are available. We have added the following text to the Methods, Results, and Discussion sections; a tornado diagram presenting the ICER variation for HPV-VIA as parameters are varied is now included in the Appendix (Figure A6).

p. 12, Methods sub-section, Scenario analysis: “The base case and sensitivity analysis assumed the availability of all strategies (i.e., Pap; HPV-Cryo; HPV-VIA; and HPV-Pap). Additionally, we performed a scenario analysis in which we assumed HPV-Cryo was not available for logistical and programmatic reasons (i.e., limited access to cryotherapy equipment and gas).”

p. 15, Results sub-section, Scenario analysis: HPV-Cryo unavailable: “When we assumed HPV-Cryo was not available as a screening strategy, HPV-VIA was the least costly and most effective strategy in the base case and most sensitivity analyses, with a base case ICER of US\$550 per YLS (Appendix). Exceptions included the following circumstances: 1) when the direct medical cost of Pap was US\$3, Pap alone had a more attractive ICER (US\$530), albeit was less effective than HPV-VIA (US\$630 per YLS); 2) Pap performance in the general screening population was improved and Pap alone became the most effective strategy, with a more attractive ICER than less effective strategies (US\$540 per YLS); 3) VIA test sensitivity in HPV-positive women was only 0.40, in which case HPV-VIA had an ICER of US\$726 per YLS, but HPV-Pap was more effective with an ICER of US\$3,260.”

p. 17, Discussion: “When we assumed a scenario in which HPV-Cryo was not available for logistic and programming reasons, we found that HPV-VIA was usually the least costly and most effective of the remaining strategies, and, although the ICER was less attractive than the ICER associated with HPV-Cryo in the main analysis, it was below Nicaragua’s per capita GDP.”

We believe the range of HPV-VIA test performance in HPV-positive women explores a sufficiently low range to demonstrate deteriorated performance (i.e., 40% sensitivity for CIN2+ in HPV-positives) in older women.

We have clarified the presentation of test performance variables presented in Table 1 (see below).

3. The definition “HPV screen-and-treat” is somehow misleading, as it suggest that HPV detection and cryotherapy for HPV-positive occurs within the same visit. “HPV-Cryo” would be less confusing.

We have made the suggested change throughout the manuscript and supplementary material.

4. Table 1, column “sensitivity analysis”. It is not clear which sensitivity and specificity ranges were explored. To avoid confusion I suggest to use the following notation “x to y” or “x - y” to denote a range.

Because we did not explore test sensitivity and specificity separately across a range (due to the correlation between the two variables along an ROC curve) but rather considered these parameters as moving together in pairs, we have clarified the Table entries to note “alternative sensitivity/specificity pairs.”

5. Page 16, lines 328-29 “individual-level data for each woman were not available; thus, our estimates represent average costs in the project populations”. Does this mean that ONLY average costs were available (as I expect) or that you had access to a distribution of the costs. In the latter case, you could in principle account for such distribution in your analysis.

Yes, generally only average costs were available for each department (with a few exceptions, such as the number of HPV specimens typically transported per trip from the clinic to the lab). The ranges for sensitivity analysis are intended to include these ranges and additional uncertainty where ranges for other parameters that impact a bundled input cost (e.g., the direct medical cost of HPV testing) are unavailable.

6. Figure 5 may be improved: a) Groups of variables are difficult to identify (subheadings should be better identifiable); b) a large fraction the plot area (from 750 USD to 2000 USD on the horizontal axis) is almost non-informative.

We have fixed the groupings of variables so that readers may more easily distinguish subheadings. We understand the concern that the area of the plot from US\$750 to US\$2,000 is blank, but we feel this space is necessary in order to display that the ICERs for HPV-Cryo consistently fall well below per capita GDP (the selected threshold for interventions that would be “very cost-effective”).

7. Refs 28 and 42 refer to the same paper, i.e. Ronco et al, Lancet 2014.

We have corrected this error.

VERSION 2 – REVIEW

REVIEWER	Talía Malagón McGill University, Canada
REVIEW RETURNED	13-Feb-2017

GENERAL COMMENTS	Great work by the authors. I only have some cosmetic comments regarding the figures and tables for more clarity: Figure 1: While the figure indicates the VIA test is done at the referral facility, the Table A6 indicates the VIA test is done at the primary facility. Which is correct? The authors should fix this discrepancy. The authors should also use the same terminology between the figure and the table (I assume primary facility=screening facility)? Figure 2: The font size for the color key and x-axis is too small. The patterns the authors have chosen are low-quality images; I would have preferred patterns similar to those used in appendix Figure A3 which are much clearer. The title should not be on the figure itself but in the legend (as per ICMJE recommendations). A bar chart may be inefficient for this purpose; the authors could consider replacing this figure with a Table instead, which allows better comparison of specific categories of costs, and would allow showing the mean cost differences as recommended by CHEERS. Figure 3: The font size for the axes should be larger. The title should not be on the figure itself but in the legend (as per ICMJE recommendations). Figure 4: The patterns the authors have chosen are low-quality images; I would have preferred patterns similar to those used in appendix Figure A3 which are much clearer. The title should not be on the figure itself but in the legend (as per ICMJE recommendations). Figure 5: The font size for the axes is too small. The figure could be redimensioned to a longer length to accommodate a larger font size on the y axis. Figure A6: The font size for the axes is too small. The sideways text
---

	on the y-axis should be rotated for left-to-right reading as the current orientation leads to broken unreadable words. The figure could be redimensioned to a longer length to accommodate a larger font size on the y axis. Table A6: The explanation is missing for the b superscript.
--	---

REVIEWER	Iacopo Baussano International Agency for Research on Cancer World Health Organization 150 cours Albert Thomas 69372 Lyon CEDEX 08, France
REVIEW RETURNED	17-Feb-2017

GENERAL COMMENTS	The authors of the paper have satisfactorily addressed all the issues and comments proposed by the reviewers. In my opinion the manuscript should be accepted.
--

VERSION 2 – AUTHOR RESPONSE

Reviewer 1: Talía Malagón

1. Great work by the authors. I only have some cosmetic comments regarding the figures and tables for more clarity.

We thank the Reviewer for the positive feedback.

2. Figure 1: While the figure indicates the VIA test is done at the referral facility, the Table A6 indicates the VIA test is done at the primary facility. Which is correct? The authors should fix this discrepancy. The authors should also use the same terminology between the figure and the table (I assume primary facility=screening facility)?

We assume that VIA triage testing takes place at the referral facility. We have corrected this in Table A6 and have used the same terminology between the figure and table to (i.e., screening facility=primary facility; referral facility).

3. Figure 2: The font size for the color key and x-axis is too small. The patterns the authors have chosen are low-quality images; I would have preferred patterns similar to those used in appendix Figure A3 which are much clearer. The title should not be on the figure itself but in the legend (as per ICMJE recommendations). A bar chart may be inefficient for this purpose; the authors could consider replacing this figure with a Table instead, which allows better comparison of specific categories of costs, and would allow showing the mean cost differences as recommended by CHEERS.

We have changed Figure 2 to Table 2.

4. Figure 3: The font size for the axes should be larger. The title should not be on the figure itself but in the legend (as per ICMJE recommendations).

We have removed the title from the figure and included it in the legend only. We have also increased the size of the font on both axes.

5. Figure 4: The patterns the authors have chosen are low-quality images; I would have preferred patterns similar to those used in appendix Figure A3 which are much clearer. The title should not be on the figure itself but in the legend (as per ICMJE recommendations).

We have altered the patterns to resemble those in appendix Figure A3 and have removed the title from the figure itself.

6. Figure 5: The font size for the axes is too small. The figure could be redimensioned to a longer length to accommodate a larger font size on the y axis.

We have redimensioned the figure to accommodate a larger font on the y-axis.

7. Figure A6: The font size for the axes is too small. The sideways text on the y-axis should be rotated for left-to-right reading as the current orientation leads to broken unreadable words. The figure could be redimensioned to a longer length to accommodate a larger font size on the y axis.

We have redimensioned the figure to accommodate a larger font on the y-axis. The sideways text has also been rotated.

8. Table A6: The explanation is missing for the b superscript.

We have removed the b superscript.

Reviewer: 3

Reviewer Name: Iacopo Baussano

1. The authors of the paper have satisfactorily addressed all the issues and comments proposed by the reviewers. In my opinion the manuscript should be accepted.

We thank the Reviewer for the positive feedback.